# SALIENCY GRAFTING: INNOCUOUS ATTRIBUTION-GUIDED MIXUP WITH CALIBRATED LABEL MIXING

## ABSTRACT

The Mixup scheme of mixing a pair of samples to create an augmented training sample has gained much attention recently for better training of neural networks. A straightforward and widely used extension is to combine Mixup and regional dropout methods: removing random patches from a sample and replacing it with the features from another sample. Albeit their simplicity and effectiveness, these methods are prone to create harmful samples due to their randomness. In recent studies, attempts to prevent such a phenomenon by selecting only the most informative features are gradually emerging. However, this maximum saliency strategy acts against their fundamental duty of sample diversification as they always deterministically select regions with maximum saliency, injecting bias into the augmented data. To address this problem, we present *Saliency Grafting*, a novel Mixup-like data augmentation method that captures the best of both ways. By stochastically sampling the features and 'grafting' them onto another sample, our method effectively generates diverse yet meaningful samples. The second ingredient of *Saliency Grafting* is to produce the label of the grafted sample by mixing the labels in a saliency-calibrated fashion, which rectifies supervision misguidance introduced by the random sampling procedure. Our experiments under CIFAR and ImageNet datasets show that our scheme outperforms the current state-of-the-art augmentation strategies not only in terms of classification accuracy, but is also superior in coping under stress conditions such as data corruption and data scarcity.

