# OpenReview forum: "Saliency Grafting: Innocuous Attribution-Guided Mixup with Calibrated Label Mixing"
_ICLR.cc/2021/Conference — Reject_

### Official Review · AnonReviewer1 · 2020-10-27
**Would like to see strong evidence of why Puzzle Mix lacks of diversity and how it negatively impacts the model’s performance**

**Rating:** 5
**Confidence:** 4

**Review:**

The paper propose a novel method to address the lack of diversity in mixed samples created by the saliency-based Mixup strategies such as Puzzle Mix. To attain this goal, the proposed approach incorporates a random process in the saliency regions selection process to give all salient regions equal chance, and accordingly adjust the modeling target to match the saliency of the mixed image. The paper presents an interesting extension to the existing work. Nevertheless, I would like to see more evidence of why the existing methods lack of diversity and how it negatively impacts the model’s performance. I also think the experiments in the current form is weak. My major comments are as follows.

Regarding the proposed method:

1.	The motivation of generating diverse mixed images through selecting diverse salient features is interesting to me. But I think the current justification is weak. It is true that the Puzzle Mix will select saliency regions with high intensity, but I wonder if the random mixing ratio \lambda in PuzzleMix would help to generate diverse mixed samples due to the random sampling process. Another factor for the diversity here is that even using similar saliency regions of the input pair for the mixed images, the mixed labels are different due to the random selection of the mixing ratio. In this sense, I think how severe this lack of diversity issue is and how it negatively impacts  the performance of the mode are not very clear to me.  It would be very beneficial if the paper could show it somehow.
2.	The way that the proposed method generates the mixed modeling target for the mixed input reminds me of this paper: nonlinear mixup: out-of-manifold data augmentation for text classification (AAAI2020). In the nonlinear Mixup method, the mixed label for a mixed input in Mixup is also computed based on the input pair, although the method is evaluated using text classification tasks. I think it would be useful to discuss the difference and adjust the content such as Table1 and Section3 to make the statements in the paper more accurate.

3.	Some parts in the proposed method deserve further discussion and justification. For example, the Threshold for the normalized saliency map. How sensitive is the threshold value to the performance of the proposed model? It would be useful to include some analysis either theoretically or experimentally.

Regarding the experiments:

1.	I am surprised to see from Table2 that the PuzzleMix degraded the performance of the baseline model. I wonder why this happened. In this sense, I would like to see experiments with some other benchmarking datasets such as Cifar10, SVHN, or TinyImagenet. Also, for these experiments, it would be useful to also provide the deviation of the 3 runs.

2.	For Cifar100 using WRN28-10 (also for ImageNet with ResNet-50), the difference between the proposed method and the PuzzleMix is very small. This suggests that it would useful to experiment on other network architectures such as ResNet-18 or ResNet-50, which are used in the original Mixup and the PuzzleMix papers.

Minor issues:

1.	The paper discusses the harmful samples potentially generated by Mixup in several places without citation. I think it would be useful to include some citations such as the manifold intrusion issue raised in the AdaMixup paper (AAAI2019) or the noise image issue as discussed in the PuzzleMix paper (ICML2020).

 In short, I think the paper would benefit from further justifying the problem of lack of mixed sample diversity in the PuzzleMix. Also, the current experimental results show minor improvement over PuzzleMix, which also makes the contribution of the paper less significant.

---

> ### Author Response · Authors · 2020-11-19
> **Response to Reviewer #1 (4/4)**
>
> ### [Q5. Experiments on other benchmarking datasets]
> To address the reviewer's concern, we evaluated our method on another benchmark dataset - TinyImageNet. We trained a ResNet-18 for 600 epochs with an SGD+momentum optimizer, following the one of TinyImageNet experimental settings in the PuzzleMix paper. Other data augmentation methods including PuzzleMix are evaluated using the released code and authors’ hyperparameters. The obtained results are shown in Table C. Like the experiments contained in the main paper, Saliency Grafting consistently exhibits the best performance on this new benchmark dataset. However, PuzzleMix falls behind CutMix.
>
> Table C. Top-1/Top-5 errors on TinyImageNet for ResNet-18 in comparison to state-of-the-art data augmentation methods.
>
> | Method | Top-1 Err. | Top-5 Err. |
> |-|:-:|:-:|
> | Saliency Grafting | **34.80 (±0.095)** | **16.13 (±0.077)** |
> | PuzzleMix | 36.67 (±0.081) | 16.92 (±0.074) |
> | CutMix | 36.13 (±0.064) | 16.20 (±0.042) |
> | Mixup | 39.01 (±0.153) | 19.56 (±0.052) |
>
> ---
> ### [Q6. CIFAR 100 experiment on other network architectures]
> In response to the reviewer`s constructive suggestion, we assess our method on CIFAR-100 using another model architecture, ResNet18. We trained the network for 300 epochs with an initial learning rate of 0.1 and decayed the learning rate at 150, 225 epoch. Other data augmentation methods including PuzzleMix are evaluated using the released code and authors’ hyperparameters. The results are displayed in Table D. In this architecture setting, Saliency Grafting again shows the best performance. However, PuzzleMix falls behind Mixup and CutMix.
>
> Table D. Top-1/Top-5 errors on CIFAR-100 for ResNet-18 in comparison to state-of-the-art data augmentation methods.
>
> | Method | Top-1 Err. | Top-5 Err. |
> |-|:-:|:-:|
> | Saliency Grafting | **20.39 (±0.074)** | **5.80 (±0.052)** |
> | PuzzleMix | 22.51 (±0.092) | 6.39 (±0.101) |
> | CutMix | 20.97 (±0.060) | 5.89 (±0.026) |
> | Mixup | 22.04 (±0.124) | 6.96 (±0.101) |
>
> ---
> ### [Q7. It would be useful to include some citations such as the manifold intrusion issue raised in the AdaMixup paper (AAAI2019)]
> Thank you for the helpful suggestion. This paper addressed the "manifold intrusion" problem, which is a mismatch problem between the mixup sample and the corresponding soft label. We agree that citing this interesting paper would be beneficial, and will add the references properly in the revision.
> ** Updated ** We've cited this novel work in our introduction section.

---

> > ### Comment · AnonReviewer1 · 2020-11-24
> > **Question regarding results on Cifar100.**
> >
> > Thanks for the additional experiments. I have a question regarding the results on Cifar100. For the ResNet-18, the PuzzleMix paper reported a Top-1 error rate of 19.62%, which differs from the result of 22.51% reported here. I wonder what was the reason for that.
> >
> > Also, the results on the Tiny ImageNet look good. I wonder if you have the results on the ResNet-50 since Puzzle seems work well on large networks.

---

> > > ### Author Response · Authors · 2020-11-25
> > > **2nd response to Reviewer #1 (3/3)**
> > >
> > > ### [Q. Reproducing results of Puzzlemix on Cifar-100]
> > > As we have reported in the response for Q4 of the initial author response, the low performance of PuzzleMix is due to PuzzleMix requiring an **abnormal** amount of epochs.
> > > The reported Cifar-100 results of the PuzzleMix paper are run for 1200 epochs (which is peculiarly long). Instead, our experiment (using the author code) results are reported for 300 epochs, which is a more **conventional experiment setting in data augmentation papers** (especially for light models such as ResNet-18) [1,2]. However, one might argue that the performance of PuzzleMix can only be observed when used in specific, paper-reported settings. To this end, we compare the methods inside **PuzzleMix’s comfort zone**.  We also conducted other experiments (WRN28-10 on CIFAR-100, ResNet-18 on TinyImageNet) according to the experimental settings of the original PuzzleMix paper and confirmed that Saliency Grafting achieves the best performance even in these cases.
> > >
> > > [1] Zhong et al., 2020. https://arxiv.org/abs/1708.04896v2
> > > [2] Yun et al., 2019. https://arxiv.org/abs/1905.04899
> > >
> > > ---
> > >
> > > ### [Q. I wonder if you have the results on the ResNet-50 since PuzzleMix seems work well on large networks.]
> > > As we’ve mentioned in the above comment, we could not conduct the Cifar-100 on ResNet-50 experiment due to time constraints. However, we possess an ImageNet classification experiment and weakly supervised object localization experiment with CUB-200-2011 on **ResNet-50** (Appendix A). We also conducted an experiment using PyramidNet-200, which is larger than the ResNet-50 architecture, where Saliency Grafting outperformed all other methods. In this context, we can state our method is  NOT restricted to a specific model architecture, and works well on a variety of model sizes and architectures.

---

> ### Author Response · Authors · 2020-11-19
> **Response to Reviewer #1 (3/4)**
>
> ### [Q3. How sensitive is the threshold value to the performance of the proposed model?]
> The threshold value of our method is determined by the expectation of the temperature - scaled saliency map (see Section 4). Note that the number of saliency regions greater than the expectation depends on the temperature $T$. As $T$ decreases, the softmax distribution becomes sharper and the number of saliency regions above the expectation decreases. That is, the mixing regions are selected from a smaller range. On the other hand, as $T$ increases, the distribution flattens so that nearly half the numbers are above the threshold. To see the sensitivity of model performance with respect to the softmax temperature, we conducted an additional experiment on the CIFAR-100 dataset with ResNet-18 by increasing the temperature from 0.01 to 0.30 (Appendix A  Figure 4). If we set a very small $T$, such as 0.01, only a small number of regions are mixed, resulting in a relatively small performance improvement. As we raise the temperature, the number of participating regions increases, resulting in a major increase in performance. When the temperature is sufficiently high, enough number of regions can participate in the mix. Thus, further increasing the temperature plateaus the performance.
>
> ---
> ### [Q4. Why does PuzzleMix exhibit degraded performance in Table 2 of the main paper? (CIFAR100 w/ PyramidNet-200 experiment) ]
> Please note that our experiment on CIFAR-100 with PyramidNet-200 in Table 2 just follows the setting of CutMix paper. This setting that trains the network for 300 epochs while decaying the learning rate at 150, 225 epochs, is widely employed in data augmentation papers [1]. The experiment for PuzzleMix was conducted using the author’s code. Also, we adopted the 4 hyperparameters (label smoothness $\beta$ , data smoothness $\gamma$, prior term $\eta$, transport cost $\xi$) as described in the PuzzleMix paper. Despite the alignments made, PuzzleMix exhibited degraded performance. However, such a phenomenon is again observed in an additional experiment (Q6), where PuzzleMix lags behind simpler augmentation methods such as CutMix and Mixup.
>
> We conjecture that the degraded performance of PuzzleMix on CIFAR-100 with PyramidNet-200 partly originates from the epoch difference (In the Puzzlemix paper, the authors trained the model for 1200 epochs for ResNet18 on CIFAR-100, which is generally a very long experiment.). Nevertheless in this experiment, running more epochs only for PuzzleMix is unfair since all other methods also tend to continuously improve as the number epochs increases. In addition, since the qualitative wall-clock training time per epoch of PuzzleMix is almost 4 times that of simple random augmentation (CutMix, Mixup) and reaches up to 2.5 times of that of Saliency Grafting, it is difficult to train a parameter-heavy model such as PyramidNet-200 with a method that requires total 16x wall clock time (4x per epoch and 4x epochs). The significant computation cost of PuzzleMix is due to the increased number of backward passes and the additional optimization problems for each iteration.
>
> Since PuzzleMix was not able to reach the expected performance, we conducted an additional experiment on PuzzleMix’s terms. In our original submission, we included a WRN28-10 experiment following the PuzzleMix paper setting. By adopting the exact setting described in the PuzzleMix paper, we can directly compare the performance of our method with the sufficiently converged, **author-reported performance of PuzzleMix (Table 10)**. In this experiment, Saliency Grafting exhibits superior performance compared to the reported performance of PuzzleMix.
> Our attempt to reproduce the author-reported values using the author codes resulted in Top-1 16.18 (+- 0.148) and Top-5 3.78 (+- 0.032). If we compare Saliency Grafting with the reproduced values, the Top-1 error gap is widened (our method is still the best).
>
> Also, as suggested by the reviewer, additional experiments were conducted on another dataset, and another architecture, which is described in the next questions Q5 and Q6.
>
> [1] https://arxiv.org/abs/1708.04896v2

---

> ### Author Response · Authors · 2020-11-19
> **Response to Reviewer #1 (2/4)**
>
> TableA. Top-1 error rates on TinyImageNet’s test set with top-k% of salient regions removed (ResNet-18)
>
> | Method | Top-1 Err. (0%) | Top-1 Err. (12.5%) | Top-1 Err.(25%) |
> |-|:-:|:-:|:-:|
> | Saliency Grafting | 34.80 (±0.095) | 38.11 (±0.019) | **48.25 (±0.247)** |
> | PuzzleMix | 36.67 (±0.081) | 49.48 (±0.099) | **72.86 (±0.210)** |
>
> TableB. Top-5 error rates on TinyImageNet’s test set with top-k% of salient regions removed (ResNet-18)
>
> | Method | Top-5 Err. (0%) | Top-5 Err. (12.5%) | Top-5 Err.(25%) |
> |-|:-:|:-:|:-:|
> | Saliency Grafting | 16.13 (±0.077) | 18.59 (±0.085) | **24.73 (±0.182)** |
> | PuzzleMix | 16.92 (±0.074) | 25.05 (±0.055) | **48.59 (±0.083)** |
>
> For further justification, we design and conduct an experiment to demonstrate that a ‘maximizing saliency strategy’ such as PuzzleMix harms model performance due to its lack of diversity. First, using a separate model pre-trained with neither PuzzleMix nor SaliencyGrafting, we produce saliency maps of the images in the test set. Using these saliency maps, we remove the top-k% (k = 12.5, 25) salient regions of each image. Using this modified test set, we evaluate the performance of two models trained with SaliencyGrafting and PuzzleMix, respectively. This experiment exhibits intriguing results (Table [A, B]), particularly in the case where the top-25% of salient regions are removed. Saliency Grafting is relatively superior at defending against performance degradations as k increases, while PuzzleMix’s error rates increase significantly. In this sense, the following insights can be obtained:
> Since the model trained with PuzzleMix is used to being provided the most salient region of an object, its classification greatly depends on this region being present. Thus, when this region is **removed** or occluded, the performance drastically decreases. However, Saliency Grafting does not have this problem, since it is possible for the thresholding & stochastic sampling step to remove these most salient regions, forcing the model to correctly classify using other parts of the image. Regional dropout methods such as CutMix and Cutout also generate such regional diversity to increase generalization performance. Therefore, even though a set of PuzzleMix/AttentiveCutMix images generated through random sampling of mixing ratios from a fixed image pair are not identical to each other, it is insufficient to argue that these methods such as PuzzleMix, Attentive CutMix can guarantee effective diversity of augmented samples. This lack of diversity issue inflicts a negative effect on the generalization performance as shown in Table [A, B]. It might be easy to explain by using the above examples -The methods based on the maximum saliency strategy will struggle to classify a panda/dog with its head occluded, which is the most salient component of the image.
>
> ---
> ### [Q2. Discuss the difference with the paper - Nonlinear mixup: Out-of-manifold data augmentation for text classification (AAAI2020)]
>
> Thank you for the helpful suggestion. We find that the Nonlinear mixup paper catches similar motivation to us. This paper focused on expanding the input space of augmented samples through the nonlinear interpolation policy. Moreover, this study proposes a novel scheme that adaptively determines the mixing coefficient of the encoded labels through the multiplication of learnable parameters $\theta$ and vectorized word embeddings of nonlinear samples.  Albeit it is an interesting work in that it alleviates the limitations of the existing linear mixup, there are the following differences from our method:
> 1. Since the spatial structure is an inherent attribution of the image, it is difficult to learn target labels with a single fully connected layer using both concatenated embeddings and learnable parameters. Saliency maps are more effective in terms of capturing spatial features in natural images or spectrograms.
> 2. We propose the saliency-based label mapping function exploiting a saliency map without the learnable parameters or encoded label sets.
> 3. We also focus on the innocuity of the synthetic data not to generate the samples that possibly mislead the network. Thresholding on a temperature-scaled saliency map could effectively alleviate this issue.
>
> This comparison of the augmentation methodologies dealing with different domains will be beneficial to improve the quality of our paper. We've cited this interesting paper in our related work section (Section 2). We will also update this discussion in the preliminaries section (Section 3) in the final revision paper.

---

> > ### Comment · AnonReviewer1 · 2020-11-24
> > **Concerns regarding your results in Tables A and B.**
> >
> > Thank you for the additional experimental results in Tables A and B and the detailed explanation on the nonlinear Mixup method. I have the following concerns regarding these results.
> >
> > Since your method adjusts the modeling target to match the saliency of the mixed image, I wonder how much of such non-degradation comes from the label modification? If there is a direct connection here, then the contribution created by the diversity is arguable.
> >
> > Also, PuzzleMix seems to work well on large network,  I wonder if you have the results on ResNet-50? It would be fine if you don't have them.

---

> > > ### Author Response · Authors · 2020-11-25
> > > **2nd response to Reviewer #1 (2/3)**
> > >
> > > Thank you for your constructive feedback. It might be the last experiment we can show in this rebuttal period, but we did our best to address the reviewer's concern.
> > >
> > > ### [Q: Since your method adjusts the modeling target to match the saliency of the mixed image, …… then the contribution created by the diversity is arguable.]
> > >
> > > To clarify the reviewer's concerns, we conduct an additional Top-k removal experiment on Saliency Grafting ***without*** calibrated label mixing. This direct comparison (Saliency Grafting w\o label mixing vs PuzzleMix) enables us to identify the difference in terms of sample diversity between the two methods. Saliency Grafting w/o label mixing shows some degree of degradation compared to Saliency Grafting with label mixing, but it still exhibits **superior performance** compared to PuzzleMix (Table A, B). On the other hand, PuzzleMix displays massive performance degradations.
> > > The stark contrast between Saliency Grafting w/o label mixing and PuzzleMix proves that stochastic sampling with thresholding alone displays sample diversity that greatly surpasses PuzzleMix. However, we believe that our calibrated label mixing and stochastic sampling are **not entirely orthogonal**, as calibrated label mixing is essential to tackle potential harms that may arise from sample diversity.
> > > Moreover, as confirmed in our additional experiments (Appendix A Figure 3), if randomly sampled $\lambda$ is indeed a factor that can effectively impose the sample diversity, then in our experiment of generating data $k$ times in a minibatch (see our reply above - “Another intriguing evidence regarding”), the performance should rise as more the augmented data are generated (k-times) from the mini-batch. However, PuzzleMix did not show such tendencies.
> > >
> > > TableA. Top-1 error rates on TinyImageNet’s test set with top-k% of salient regions removed (ResNet-18)
> > >
> > > | Method | Top-1 Err. (0%) | Top-1 Err. (12.5%) | Top-1 Err.(25%) |
> > > |-|:-:|:-:|:-:|
> > > | Saliency Grafting | 34.80 (±0.095) | 38.11 (±0.019) | **48.25 (±0.247)** |
> > > | *Saliency Grafting (w/o Calibrated label)* | 35.95 (±0.212)  | 40.16 (±0.136) | **50.81 (±0.363)** |
> > > | PuzzleMix | 36.67 (±0.081) | 49.48 (±0.099) | **72.86 (±0.210)** |
> > >
> > > TableB. Top-5 error rates on TinyImageNet’s test set with top-k% of salient regions removed (ResNet-18)
> > >
> > > | Method | Top-5 Err. (0%) | Top-5 Err. (12.5%) | Top-5 Err.(25%) |
> > > |-|:-:|:-:|:-:|
> > > | Saliency Grafting | 16.13 (±0.077) | 18.59 (±0.085) | **24.73 (±0.182)** |
> > > | *Saliency Grafting (w/o Calibrated label)* | 16.62 (±0.078)  | 20.11 (±0.171) | **28.20 (±0.262)** |
> > > | PuzzleMix | 16.92 (±0.074) | 25.05 (±0.055) | **48.59 (±0.083)** |
> > >
> > > ---
> > >
> > >
> > > ### [Q: Also, PuzzleMix seems to work well on large network, I wonder if you have the results on ResNet-50? It would be fine if you don't have them.]
> > > Unfortunately, we do not have the results of Cifar-100 on ResNet-50. Due to the remaining time of the discussion phase, we could not conduct this experiment. However, we did report an experiment using ResNet-50, which is on ImageNet (Table 3). In this experiment, PuzzleMix works well(in fact, the numbers are taken from the PuzzleMix paper). Regardless, **Saliency Grafting outperformed PuzzleMix and all other methods**. Moreover, we reported an experiment on an even more complex architecture, PyramidNet-200. It is worth noting that PyramidNet-200 **(26.8M)** network is a larger network than ResNet-50 (23.5M). In the PyramidNet-200 experiment (Table 2), **our method outperformed** PuzzleMix and all other methods. With these experiments, we confirmed that our method exhibits the best performance not only with the ResNet-based architectures but also with other large architectures.

---

> ### Author Response · Authors · 2020-11-19
> **Response to Reviewer #1 (1/4)**
>
> Thank you for your thoughtful and constructive feedback.
>
> ### [Q1. Evidence of why PuzzleMix lacks diversity and how it negatively impacts the model’s performance]
> We believe that the diversity only from the randomness of mixing ratio $\lambda$ for PuzzleMix would not suffice and does not change much the fact that PuzzleMix is designed to always contain the regions with the highest saliency scores. Example images visualizing this point is shown in Figure 6 in Appendix D. In the first row of Figure 6, we are trying to mix an image of a panda image with an image of a dog using PuzzleMix. This maximum saliency strategy results in always containing the region of highest saliency, regardless of \lambda‘s value. That is, the head of the panda or the head of the dog are **always** contained in the sample created by PuzzleMix, regardless of $\lambda$. This lack of diversity (bias) results in the model overly relying on these specific regions to perform classification (overfitting). However, Saliency Grafting does not have this bias due to thresholding & stochastic sampling. The 2nd row of Figure 6 shows the samples created by Saliency Grafting. We can observe that there exist samples that the head of the panda or the dog is occluded (more diversity).
> Q3 on the threshold for normalized map below, in fact, can be a quick evidence that the diversity beyond the randomness of lambda helps to improve performance. If this threshold is given large and hence only the most salient part is selected deterministically, our method is also reduced to a mixup method that simply maximizes the saliency, as in previous studies, and experiences the performance degradation (See Q3 for more details below). Please also note that we also discussed the risk of deterministic patch selection in the ablation study (Section 5.3).

---

> > ### Comment · AnonReviewer1 · 2020-11-24
> > **noisy inputs from your approach?**
> >
> > I agree that PuzzleMix always contains the region of highest saliency, but I think the \Lambda does help to constrain how much of such region should be included in the mixed input, which may be considered as a way to generate the diversity. In this sense, I am still not convinced by your argument here.
> >
> > Also, when I look at the visualization in Appendix D, I wonder if your method would generate noisy inputs (when comparing to PuzzleMix). If you look at the images created by your method in the middle column, it seems the mixed images may contain noise features. For example, for the Bird and Bear, the Bird features used in the mixed input could be very noisy. Similar issue can be seen at the bottom row for the Shark features.

---

> > > ### Author Response · Authors · 2020-11-25
> > > **2nd response to Reviewer #1 (1/3)**
> > >
> > > We sincerely appreciate your effort to review our paper. Thanks for the constructive comments. We have responded to each of your three comments.
> > >
> > > ### [Q: I agree that PuzzleMix always contains the region ...... considered as a way to generate the diversity.]
> > >
> > > It is true that $\lambda$ determines how much salient region should be included, generating diversity. However, the amount of diversity generated thus is **not enough**, as randomly sampling the $\lambda$ only results in expansions/contractions of the region with the maximum saliency patch inside.
> > > - The risk of selecting the highest saliency is well evaluated in our top-k removal experiment (Table A and B).
> > > - Please also check our new experimental result in Appendix A Figure 3 and our response above where we produce $k$ independent augmented instances with different samples of $\lambda$. Here, the performance of Saliency Grafting consistently improves as k increases, whereas PuzzleMix is predisposed to maintain somewhat constant performance even when k increases.
> > >
> > > To summarize, sampling $\lambda$ does grant some diversity to PuzzleMix, but such diversity is **relatively small**, especially in the sense of **diversity of sample difficulty**, as PuzzleMix can only generate **easy** samples (maximum saliency region always included), while Saliency Grafting can generate samples of **diverse difficulties** (as in excluding maximum saliency region), resulting in superior generalization performance.
> > >
> > > ---
> > >
> > >
> > > ### [Q: Also, when I look at the visualization in Appendix D, I wonder if your method would generate noisy inputs (when comparing to PuzzleMix). …… Similar issues can be seen at the bottom row for the Shark features.]
> > >
> > > We are not sure about these “noises” the reviewer addresses. We think that it might refer to:
> > >
> > > 1. non-object regions (background regions) being pasted on top of object regions
> > > 2. object regions being pasted on top of non-object regions
> > > 3. object regions being pasted on top of object regions
> > > 4. grafting (occlusion) patterns that look “noisy” to humans (e.g. checkerboard-like pattern).
> > >
> > > For *case 1*, these kinds of samples are beneficial to the generalization of the model since there exist many cases where the target object is occluded by an obstacle(e.g. only the rear end of a car is shown as the front is blocked by a building). The benefits of such samples are evident in Cutout [1] and Random Erasing [2]. Also, unlike other random augmentation, grafting only the background region is hard to occur in our method that exploits a temperature-scaled saliency map.
> > > For *case 2*, when an object region is grafted on top of a non-object region, 2 objects exist in a single sample. These kinds of samples are frequently generated in PuzzleMix and benefit model training.
> > > For *case 3*, when an object region is grafted on top of an object region, the image becomes a harmful sample(in a certain sense, noisy) as the overwriting image may cause label mismatch/miscalibration. However, we **rectify** this noise by using our calibrated label mixing.
> > > For *case 4*, keep in mind that we are discussing a data augmentation scheme. These cases are not detrimental to training, but beneficial. The reviewer suggested that the bird-dog and shark-rabbit images are “noisy”. Yet, by looking at the images, we humans can *still distinguish* all objects(bird/dog/shark/rabbit). Thus, this “noise” makes the sample a hard one, but not a wrong one (especially with calibrated label mixing). By feeding these hard samples to the model, the model is prepared to classify such test-phase images with difficult real-life occlusion patterns. In this sense, we believe that providing samples of a variety of difficulties is beneficial to the generalization performance of the model.
> > >
> > > [1] Terrance DeVries and Graham W Taylor. Improved regularization of convolutional neural networks with cutout. arXiv preprint arXiv:1708.04552, 2017.
> > >
> > > [2] Zhun Zhong, Liang Zheng, Guoliang Kang, Shaozi Li, and Yi Yang. Random erasing data augmentation. In Proceedings of the AAAI Conference on Artificial Intelligence (AAAI), 2020.

---

> ### Author Response · Authors · 2020-11-23
> **Another intriguing evidence regarding the diversity of augmented data (Newly Updated)**
>
> ### [Evidence of why PuzzleMix lacks diversity and how it negatively impacts the model’s performance]
> To further verify our claim, we conducted another intuitive experiment to compare Saliency Grafting and PuzzleMix in terms of sample diversity. In this experiment, for every iteration, each method trains the network by generating additional augmented data $k$ times from the mini-batch; each method produces $k$ independent augmented instances with its randomness. In order to ensure sufficient diversity, the mixing ratio $\lambda$ is also newly sampled for each augmented data. While varying $k$ from 1 to 6, we evaluate whether each method can obtain the performance gain due to sample diversity. We follow Puzzlemix's WRN28-10 training setting for 200 epochs, and use 20% of the Cifar-100 dataset to better confirm the diversity effect of the augmented data (and partially due to the time constraint of rebuttal period). The average error rate for 5 random seeds is reported. (Please see the experimental result in Appendix A Figure 3)
>
> As shown in Figure 3, the performance of Saliency Grafting consistently improves as k increases, whereas PuzzleMix is predisposed to maintain somewhat constant performance even when k increases. In this sense, we believe that this is the direct evidence that generating PuzzleMix’s samples by sampling the random mixing ratio is insufficient to ensure sample diversity. However, since Saliency Grafting exploits temperature-scaled thresholding with stochastic sampling, the model easily attends the entire object as $k$ increases. Also, it is possible to properly supervise the augmented data through calibrated label mixing, sample diversity can be guaranteed innocuity.

---

### Official Review · AnonReviewer3 · 2020-10-28
**borderline paper**

**Rating:** 6
**Confidence:** 4

**Review:**

This paper advances the line of research in data augmentation following Mixup paper. Cutmix is a image-specific variant of mixup that pastes a rectangular region from a donor image to a target image; however, it does this in a completely random fashion not paying attention to whether the discriminative parts of either image are retained. Recent methods alleviate this problem by using saliency guided region selection either preserving the most discriminative parts of the donor image (Attentive Cutmix) or both images (PuzzleMix). However, in doing so they replace the random nature of Cutmix with a deterministic approach potentially reducing sample diversity. The first contribution of the paper is a saliency guide stochastic augmentation method which combines the best of both worlds. The second contribution is in the generation of more meaningful labels for the newly created image by taking into account the saliency of the regions used from both images rather than blindly using the mixing ratio. The contributions of the paper are somewhat incremental but are reasonable. The validity of using saliency in determining the label of the new sample is clear. The experimental results are mixed. Most experiments demonstrate minor improvements over previous methods. On the other hand the top-5 performance of CutMix is better than the proposed method in Table 2. With such small differences and no standard deviations on accuracy reported it is not clear whether the improvements are really significant.

In the data scarcity experiments, I find the statement “ Note that the performance of CutMix deteriorates as the number of data per class decreases due to their randomness occurring
label mismatching.” misleading. This is true of every method in that table not just CutMix, and the detoriation amounts are very similar.

Clarity: Overall I found the paper well written and easy to read.

The authors have provided standard errors which improves the confidence that the improvements in the experiments are significant.

---

> ### Author Response · Authors · 2020-11-19
> **Response to Reviewer #3 (2/2)**
>
> ### [Q2: Most experiments demonstrate minor improvements over previous methods. With such small differences and no standard deviations on accuracy reported it is not clear whether the improvements are really significant.]
>
> Thank you for your feedback. We report the standard errors of CIFAR-100 for PyramidNet-200 (Table 2 in the main paper) in Table A (below). Also, in addition to the experiments in the main paper, we conducted 2 additional experiments to benchmark Saliency Grafting: TinyImageNet for ResNet-18 (Table B) and CIFAR100 for ResNet-18 (Table C), and also report standard errors for both of them. In these experiments, Saliency Grafting *consistently* outperforms other methods. This performance gap is a major gain for a data augmentation method, as the gain is achieved not by changing architectural designs, which are the main powerhouse of performance, but by tweaking data processing schemes, which are small but effective finishing touches. We also observe that for all of these experiments, the standard errors are far from overlapping, conveying statistical significance.
> In the main paper, we omitted standard errors to align our experiments with previous papers, as they have not reported any standard errors. However, we agree with R3 that standard errors are necessary. Since the results reported in previous papers do not contain standard errors, we reproduced them using author codes and guidelines for repetition. However, for some methods, we were unable to reproduce the performance stated in their respective papers, albeit using the author’s code/hyperparameters and following their guidelines.
>
> Table A. Top-1/Top-5 errors on CIFAR-100 for PyramidNet-200 in comparison to state-of-the-art data augmentation methods.
>
> | Method | Top-1 Err. | Top-5 Err. |
> |-|:-:|:-:|
> | Saliency Grafting | **13.59 (±0.062)** | **3.01 (±0.021)** |
> | PuzzleMix | 16.52 (±0.066) | 3.70 (±0.065) |
> | CutMix | 14.51 (±0.149) | 3.08 (±0.024) |
> | Attentive CutMix | 15.24 (±0.085) | 3.46 (±0.061) |
>
> Table B. Top-1/Top-5 errors on TinyImageNet for ResNet-18 in comparison to state-of-the-art data augmentation methods.
>
> | Method | Top-1 Err. | Top-5 Err. |
> |-|:-:|:-:|
> | Saliency Grafting | **34.80 (±0.095)** | **16.13 (±0.077)** |
> | PuzzleMix | 36.67 (±0.081) | 16.92 (±0.074) |
> | CutMix | 36.13 (±0.064) | 16.20 (±0.042) |
> | Mixup | 39.01 (±0.153) | 19.56 (±0.052) |
>
> Table C. Top-1/Top-5 errors on CIFAR-100 for ResNet-18 in comparison to state-of-the-art data augmentation methods.
>
> | Method | Top-1 Err. | Top-5 Err. |
> |-|:-:|:-:|
> | Saliency Grafting | **20.39 (±0.074)** | **5.80 (±0.052)** |
> | PuzzleMix | 22.51 (±0.092) | 6.39 (±0.101) |
> | CutMix | 20.97 (±0.060) | 5.89 (±0.026) |
> | Mixup | 22.04 (±0.124) | 6.96 (±0.101) |
>
> ---
>
> ### [Q3: Not only CutMix but all methods struggle against data scarcity.]
> The reviewer misunderstood our intentions. Indeed, the performances of all methods degrade as the number of data per class decreases. However, the meaning we tried to convey was that CutMix suffers the most from data scarcity. In Table 4 of the main paper, we reported results (top-1 error on CIFAR-100) for 3 data portions: 10%, 20%, and 50%. When data scarcity is not severe (50%), CutMix (28.73) performs slightly better than Mixup (28.97) and is relatively close to PuzzleMix (28.04). However, as the amount of data decreases, the performance of CutMix degrades rapidly, displaying the worst performance (lags behind all augmentation methods) among the augmentation methods as we hit 20% and 10% data (On 10% data, the top-1 error of CutMix is 55.71 while Mixup is 51.86 and PuzzleMix is 52.63). We will clarify the statement in the revision.
>
> We suspect that this is because CutMix carries an innate flaw that is **label mismatch**, due to the saliency-agnostic nature of its scheme. A previous study shows that label noise is more destructive when data is scarce (Rolnick et al. 2017 [1]). We suspect that this is why CutMix performs worse in data scarcity situations.
> PuzzleMix and Mixup incompletely address this problem by generating incorrectly calibrated labels. However, our method produces well-calibrated (correct) labels which are shown to be more resistant to data scarcity, with our method displaying the best performance.
>
> [1] https://arxiv.org/abs/1705.10694

---

> > ### Author Response · Authors · 2020-11-24
> > **Thank you for checking out our response. Please let us know of any unasked questions.**
> >
> > Thank you for checking out our response. We have resolved the questions raised in the reviewer feedback: the significance of our findings, the validity of our experiments, and clarifications regarding data scarcity experiments. If there were any unasked questions or ambiguations that have contributed to the score but were omitted in the initial review, please let us know.

---

> ### Author Response · Authors · 2020-11-19
> **Response to Reviewer #3 (1/2)**
>
> Thank you for your thoughtful and constructive feedback. To dispel your concerns about the performance improvement, we report the standard errors of CIFAR-100 for PyramidNet-200 (Table 2 in the main paper) in Table A (below), and evaluate our method with two additional benchmark experiments: TinyImageNet for ResNet-18 (Table B) and CIFAR100 for ResNet-18 (Table C), both with standard errors. Since Saliency Grafting consistently outperforms other methods with non-overlapping standard errors, we believe that the consistent excellence of Saliency Grafting is not minor nor coincidental.
>
> ### [Q1: Top-5 performance of CutMix (CIFAR-100 PyramidNet) is better than the proposed method.]
>
> In fact, we tried to reproduce the CutMix results by following all guidelines from the author including official code and hyperparameters, but we only achieved the value of **3.08 (±0.024)** (Top-5 err) for the CIFAR-100 PyramidNet case. According to this result, **Saliency Grafting outperforms CutMix** without overlapping the standard errors (see Table A of Q2). In our original submission, however, we respected the values reported in CutMix.
>
> Nevertheless, CutMix works quite well compared to other baselines in top-5 accuracy, and we make the following conjectures. One of the core components of our method is **calibrated label mixing**. The benefits provided by this component is best felt in top-1 accuracy. CutMix, which does not perform calibrated label mixing, generates 2-hot incorrectly calibrated label signals. A model trained with this signal will be weak in terms of top-1 accuracy since, for a portion of augmented samples, the top-1 and top-2 labels would be reversed. However, in terms of top-5 accuracy, the effect of this incorrectness is shrunk since both top1 and top2 are within the top-5 threshold.

---

### Official Review · AnonReviewer2 · 2020-11-03
**Well written. Clear and effective method. State-of-the art results with thorough ablation study.**

**Rating:** 7
**Confidence:** 4

**Review:**

The authors propose a novel sailency-guided data augmentation method that alleviates some of the drawbacks arising with recent Mixup-based augmentation approaches.  Specifically, the authors propose sailency thresholding for region selection (instead of maximum sailent region), stochastic sampling of sailent patches, and sailency-based label mixing. The results show clear improvements over competitors, and an ablation study shows the performance gains of each of the design choices discussed previously.

The paper is pleasant to read and all decissions are properly motivated. The main contributions are fairly novel, the formulation sound and the experiments well designed.

The claim of robustness of data corruption of the model seems like an over-statement. Even though the authors show that Sailency Grafting performs well under data corruption, one may attribute the robustness feature to the AugMix method that the grafting acts upon. The results of table 5. just show that grafting sailent corrupted image patches improves over just training on corrupted patches, which is something to expect given the previously reported improvements.

Early in the paper the authors claim a difference wrt. previous work that is the uniform thresholding of the salient region rather than selecting the maximum, to mitigate selection bias. I missed a section in the ablation study demonstrating the specific improvement of this decision. Also the authors briefly mention the approach to select the thresholding as the mean of the normalized sailency map. I'm curious to know if any other method of threshold selection (or patch sampling) has been explored.

---

> ### Author Response · Authors · 2020-11-19
> **Response to Reviewer #2**
>
> Thank you for your thoughtful and constructive feedback.
>
> ### [Q1: Claiming robustness (Table 5 of the main paper) is an overstatement (it is due to AugMix). It just shows that grafting salient corrupted image patches improves over just training on corrupted patches, which is something to expect given the previously reported improvements.]
> We conducted additional experiments to measure the performance of various methods without AugMix. In this setting, results show that all methods exhibit lackluster performance, including Saliency Grafting. In this sense, we agree with R1 that our method cannot guarantee significant robustness on its own. However, the intention behind our statement is that our method can be used on top of AugMix with no repercussions while generating positive synergy. Augmix, on its own, provides noise robustness but shows lackluster performance when it comes to boosting performance on clean data. When Saliency Grafting is used alongside Augmix, both clean data performance and corrupted data performance is increased beyond the numbers obtained by applying them individually (synergy). We will properly update our statements in the revision.
>
> ---
>
> ### [Q2: Ablation study for employing uniform thresholding of the salient region rather than selecting the maximum, to mitigate selection bias.]
> Please see Section 5.3 (Ablation Study) in our original submission where we presented the comparison of deterministic selection VS stochastic selection. In this section, we take AttentiveCutMix, which employs a deterministic strategy of selecting top-k pixels from a saliency map. While keeping everything else identical, we change this deterministic selection strategy of AttentiveCutMix to our stochastic selection strategy(thresholding + stochastic sampling). To keep the number of selected pixels approximately equal, we calibrated the softmax temperature so that ***k*** pixels are selected on average. Results in Table 6 show that **Stochastic+area labels** (2nd entry) show better performance compared to **Deterministic+area labels** (1st entry).
>
> ---
>
> ### [Q3: Curious to know if any other method of threshold selection (or patch sampling) has been explored?]
> We considered that it was better to select the threshold value using data statistics rather than thresholding by a fixed constant, so we adopted the expectation(mean). It is possible to adjust the number of saliency regions with a value larger than the expectation through temperature. It also is possible to control the number of saliency regions with higher values than the expectation by adjusting the shape of the softmax distribution through temperature $T$. Moreover, we plan to conduct an experiment that tweaks the $\alpha$ of the beta distribution, which is the current patch sampling distribution, to evaluate the performance difference by making the beta distribution shape concave or convex. We sincerely thank you for the helpful suggestion.

---

### Decision · Program_Chairs · 2021-01-07
**Final Decision**

**Decision:**

Reject

**Comment:**

Overall, this paper has been on the very borderline. All reviewers agree that the motivation and the idea of the paper are reasonable (although somewhat incremental) and make an interesting extension of mixup-type data augmentation. However, one expert reviewer raised some concerns which are unfortunately not fully resolved despite the intensive interaction.

The first one is the issue of diversity claimed in the paper. The authors explanation is mostly qualitative, and I, as an AC, also felt a logical jumping here, although I do understand that the proposed method somehow results in better generalization ability w.r.t. the number of generated samples as shown in Fig.3 in the appendix. This point is important because if the better generalization is coming from the label modification rather than the diversity in image space, then the novelty of the paper is limited.
Another concern is that there are some inconsistencies in the scores of previous methods implemented by the authors and in the original papers. We understand that the exact score is not always easy to reproduce, but at least it is desirable to follow the original setting (such as the number of epochs) for each method as far as possible, because the accuracy should be the most important criterion as the goal of data augmentation is better generalization. Then, authors may separately discuss computational efficiency.

Based on the discussion with reviewers as above, I conclude that the paper should be further polished and completed before publication. Thus I recommend rejection for this time.